# Optimizing learning outcomes in physical education: A comprehensive systematic review of hybrid pedagogical models integrated with the Sport Education Model

**Junlong Zhang**[1]*, **Kim Geok Soh**[1]*, **Xiaorong Bai**[2], **Mohd Ashraff Mohd Anuar**[3], **Wensheng Xiao**[2]

**1** Department of Sports Studies, Faculty of Education Studies, University Putra Malaysia, Seri Kembangan, Malaysia, **2** School of Physical Education, Huzhou University, Huzhou, China, **3** Department of Professional Development and Continuing Education, Faculty of Education Studies, University Putra Malaysia, Seri Kembangan, Malaysia

* gs62618@student.upm.edu.my (JZ); kims@upm.edu.my (KGS)

## Abstract

### Background

There is a notable gap in systematic reviews concerning hybrid pedagogical models (PMs) integrated with the Sport Education Model (SEM) and their impact on students' outcomes.

### Purpose

Which hybrid PMs incorporating SEM are currently the mainstream choices in research, and what are the main factors supporting their integration? How does SEM function as a foundational model in these hybrid teaching approaches? What learning outcomes are optimized through the hybrid models that combine SEM with other PMs?

### Methods

A systematic search was conducted in major databases in December 2023 following PRISMA guidelines. Out of the identified 1342 studies, 30 met the eligibility criteria, all of which were deemed to be of high quality.

### Results

Seven hybrid types were identified, primarily composed of two PMs, among which the blend of SEM and Teaching Games for Understanding (TGfU) emerges as the mainstream in current research. SEM, serving as the foundational structure, provides a stable framework for the hybrid, termed the "SEM + 1 model," yielding positive effects on enhancing students' learning outcomes.

**Data Availability Statement:** All relevant data are within the manuscript and its Supporting information files.

**Funding:** The author(s) received no specific funding for this work.

**Competing interests:** The authors have declared that no competing interests exist.

## Conclusions

Pedagogical models align with PMs' motivational aspects, thus enhancing learning outcomes. However, evidence for partial hybrids is lacking. Future research should explore diverse interventions, addressing coherence and teacher competence, while maintaining fidelity.

## 1. Introduction

The Pedagogical Model (PM) is a blueprint that "describes the specific procedures for organizing content, task structures, and the sequence of learning activities" [1]. Each model has specific design specifications and is considered a framework [2], allowing teachers to choose the most effective methods for delivering the model in different local contexts based on their perceived relevant teaching skills [3]. Furthermore, Pedagogical Models undergo a process of generation, testing, refinement, and further testing in various situations, reflecting their continuous modification and development [3]. Consequently, physical education (PE) has been immersed in a process of continually updating the teaching methods and PMs used in the classroom [4].

Research on teaching methods in PE encompasses various PMs that allow students to acquire breadth and depth of PE knowledge in diverse contexts [3]. As an alternative to direct instruction, a range of pedagogical models (PMs) have been introduced, including second-generation models such as Teaching Games for Understanding (TGfU) by Bunker and Thorpe [5], the Sport Education Model (SEM) by Siedentop [6], Cooperative Learning, and the Teaching Personal and Social Responsibility (TPSR) model by Hellison [7]. These PMs drive the shift from teacher-centered to student-centered instructional approaches, featuring key design characteristics that enhance not only student learning but also motivation [8]. For instance, the Sport Education Model (SEM) aims to provide students with authentic and educationally meaningful movement experiences within the school sports context, with six key structural characteristics: seasons, affiliation, formal competition, culminating activities, record-keeping, and festivals [9]. Cooperative Learning (CL) emphasizes learning together, from each other, and for each other, aiming to foster five essential elements: interpersonal skills, processing, positive interdependence, interaction promotion, and individual responsibility [10]. The core idea behind Teaching Games for Understanding (TGfU) is to shift the focus from technical aspects of gameplay to situations (tactical considerations) through modification, representation, and exaggeration [11]. It centers on placing learners in game situations where tactics, decision-making, and problem-solving are non-negotiable features, while skill practice is used to correct habits or reinforce skills. TGfU is based on six structural steps: games, game appreciation, tactical awareness, making appropriate decisions, skill execution, and performance [12]. The original TGfU model has evolved into several frameworks worldwide, such as the Step-Game Approach (SGA), Invasion Games Competence Model (IGCM), Tactical Game Approach (TGA) [13], Game Sense (GS) [14], Play Practice (PP) [15], Tactical Decision Learning Model (T-DLM) [16], and Developmental Game Stages Model (DGSM) [17], categorizing them under the umbrella term "game-centered approaches" (GCA). Health-Based Physical Education (HBPE) encourages healthy lifestyles by integrating health education into physical education programs, aiming to enhance students' physical, mental, and social well-being [18]. Social and Emotional Learning (SEL) focuses on developing students' social and emotional skills, such as self-awareness, self-management, social awareness, relationship skills, and

responsible decision-making, which are crucial for success in school and life [19]. Self-Regulated Physical Education Model (SPRM) promotes students' ability to manage their own learning processes, including goal setting, self-monitoring, and self-reflection, to become autonomous and motivated learners [20]. Finally, Teaching Personal and Social Responsibility (TPSR) is a model that focuses on fostering individual and social responsibility through five developmental goals: (1) Respect and self-control; (2) Participation and effort; (3) Self-direction; (4) Leadership and caring; and (5) Transfer. By addressing these goals, TPSR enhances students' basic psychological needs, thereby increasing their motivation and engagement in physical activities beyond PE classes [21].

Currently, the aforementioned PMs have the potential to be beneficial for students' development. However, it is widely acknowledged that no single model can be universally applicable to all PE environments [2, 22, 23]. Each model has its own strengths and limitations, and their effectiveness can vary depending on the context in which they are implemented. This is because each model is developed for specific curriculum objectives, and thus, each model has its limitations when implemented in isolation [17]. For example, while SEM aims to cultivate competent, literate, and enthusiastic individuals, it might not simultaneously foster social skills improved through Cooperative Learning (CL) group tasks or a sense of responsibility in students enhanced through Teaching Personal and Social Responsibility (TPSR) instruction. Moreover, SEM can sometimes be limited by its focus on sports and competition, which may not appeal to all students and could potentially marginalize those less interested in competitive sports [1]. Another limitation is that SEM requires significant time and resources to implement effectively, which may not always be feasible in all school contexts [2]. Additionally, SEM's structured seasons and formal competitions may not leave enough room for spontaneous or student-led activities, potentially reducing opportunities for creativity and flexibility in learning [22]. These limitations provide compelling reasons for hybridizing SEM with other PMs. Hybrid models can address the shortcomings of individual PMs by combining their strengths. For instance, integrating SEM with CL can enhance social interaction and teamwork, while blending SEM with TPSR can promote personal and social responsibility alongside sports competence. Evaluating the effects of hybrid PMs, which include SEM, is crucial to understanding their potential benefits in diverse educational settings and to developing more comprehensive and effective physical education programs [23–25].

In pursuit of maximizing benefits for students in teaching or achieving specific teaching goals, some researchers have attempted to introduce how to integrate two PMs to create a new hybrid PM [26–29]. Hybridizing PMs involves extracting and combining key features from multiple models or using one model as a base and incorporating additional essential elements from another model. For instance, in SE-TGfU, the SEM provides key organizational features (seasons, persisting teams, formal competition, record keeping, festivity, and a culminating event), while the Teaching Games for Understanding (TGfU) contributes the primary instructional approach (teacher-mediated formats (e.g., active teaching and teaching through questioning) and student-mediated formats (e.g., peer tutoring, small-group/cooperative learning approaches) to assist educators in identifying key tactical issues within each team, such as (a) tactical or strategic awareness (What do you do to keep the ball away from other players?), (b) skill execution (How do you keep the ball away from other players?), (c) time (When is the best time to pass?), (d) space (Where should you move when your teammate is trying to restart play?), (e) risk (What options do you have if your defender is near?), and (f) rationale (Why should you move after passing?) [30]. In addition, Casey and Dyson combined the emphasis on encouraging adaptation to individual differences, fair teaching, and personal and social development of CL with the TGfU, which prioritizes the development of correct decision-making and tactical awareness over skilled performance. They found that hybridizing these

two PMs is an extremely complex and challenging task, with potential benefits for both students and teachers if the models have similar characteristics and goals [31]. Likewise, recent research has demonstrated the feasibility of hybridizing TPSR and SEM by observing that they share similar learning theories and complementary objectives. For example, both models emphasize student-centered learning and personal development [32]. Combining TPSR's focus on personal and social responsibility with SEM's structure and organization can enhance student engagement and learning outcomes [32]. Moreover, hybrid models can address the limitations of individual PMs by leveraging their strengths, leading to more comprehensive educational experiences [25].

Previous studies also indicated that hybridization can enhance game performance and motor skills [26, 29] and yield positive psychosocial outcomes, such as enjoyment, willingness for physical activity, and responsibility [33, 34]. Hybrid PMs serve as effective resources for developing flexible physical education programs within multimodal projects. Teachers can choose and blend appropriate PMs based on students' situations to meet the diverse needs of different teaching objectives. Therefore, this is considered an innovative trend and a necessary means for achieving higher-quality learning outcomes [22, 35].

Despite an increasing number of studies on hybrid PMs in recent years, systematic reviews regarding the hybridization of PMs remain scarce. Indeed, only two systematic reviews have comprehensively examined various hybridizations between 2019 and 2022 [3, 35]. The first review in this field by González-Víllora et al. primarily provides fundamental information on PM hybridization in PE, covering aspects such as Hybridization, Research Focus, Participants and Context, Sport/Content, Length of Implementation, Data Sources and Analysis, and Outcomes [35]. It acknowledges that hybridization benefits students in terms of game-related skills and psychosocial variables. To extend the work of González-Villora et al., Shen & Shao further investigate the impact and mechanisms of hybrid models on students' learning outcomes (i.e., motor, cognitive, emotional, and social) and provide a comprehensive review of empirical studies on various PM hybridizations. The results suggest that the duration of implementation and teacher familiarity are the primary limiting factors for the implementation of hybrid PM teaching [3]. Notably, from the two systematic reviews, it is evident that SEM is the most frequently appearing model in hybrid PM teaching, constituting 90% and 88% in the first and second reviews, respectively. This raises the questions: which hybrid pedagogical models incorporating SEM are currently the mainstream choices in research, and what are the main factors supporting their integration? How does SEM function as a foundational model in these hybrid teaching approaches? What learning outcomes are optimized through the hybrid models that combine SEM with other pedagogical models? However, these aspects have not been extensively explored or explained in previous research. Therefore, building on prior work, this study aims to conduct a comprehensive systematic review of the role of hybrid PMs, combined with the SEM, in optimizing learning outcomes in physical education. Specifically, this study will investigate the impact of these hybrid models, integrated with SEM, on students' learning outcomes. Additionally, the study seeks to explore the advantages of SEM as a foundational model in hybrid PMs.

## 2. Methods

### 2.1 Protocol and registration

The PRISMA statement was followed in reporting this systematic review and meta-analysis [36], and the review protocol has been registered on Inplasy.com: [INPLASY202410027].

## 2.2 Data sources and search strategy

On December 3, 2024, a search of four electronic databases, namely Web of Science, EBSCO host, PubMed, and SCOPUS, was conducted to identify relevant articles on the topic. Previous reviews [3, 35] were consulted to guide the formulation of the search strategy, with keywords and Boolean operators considered both individually and in combination during the search across the four databases (see S1 Table). The search employed terms and operators such as "Sport Education," "pedagogical model," "curriculum model," "instructional models," "physical education," "hybrid*," "Integrate*," and "combine*." Additionally, a manual search of Google Scholar was performed to retrieve missing studies, including article citations and free-text searches. Moreover, all identified articles underwent reference list screening to identify any publications not initially found during the database searches (See S1 Checklist for details).

   To ensure the robustness of the data collection process, experienced librarian Miss Chen assisted. To mitigate selection bias, two knowledgeable authors (J.Z. and M.A.), both familiar with SEM, independently screened and selected studies. In cases of disagreement, a third reviewer (K.G.S) was involved to reach a consensus. Following the review of titles and abstracts, publications not meeting the inclusion criteria were excluded. Full-text examination was conducted based on exclusion criteria, and the final set of 30 articles was included for systematic review and analysis (Fig 1 and S1 File).

## 2.3 Eligibility criteria

A PICOS framework [18] was used to rate studies for eligibility. The criteria include the following:

Inclusion Criteria: (1) Published in peer-reviewed international journals; (2) Implementation of a hybrid PM incorporating the SEM in physical education (PE) environment; (3) Primary findings report at least one aspect of the impact of the hybrid PMs on student learning outcomes; (4) Articles published and written in English; (5) Empirical studies utilizing quantitative, qualitative, or mixed research methods.

Exclusion Criteria: (1) Books, book chapters, conference proceedings, master's theses, and doctoral dissertations not subjected to independent peer review are excluded; (2) Studies published in non-peer-reviewed journals and/or not indexed in the Journal Citation Reports (JCR) or Science Journal Rankings (SJR) are excluded; (3) To align with the research objectives, studies that do not design hybrid PMs with SEM and those not specifically measuring any aspect of learning outcomes are excluded.

## 2.4 Data extraction

Based on the pertinent reviews in the realm of the physical education pedagogical model [3, 35], we have succinctly outlined the distinctive features of each retained study: Sources, Purpose, Sample Characteristics, Hybrid, Length of Unit/Content, Data Collection, Study Design & Analysis, Learning Outcomes (Table 1). This process involved two reviewers (J.Z. and M.A.) obtaining information on each study through Microsoft Excel spreadsheets (Microsoft Corporation, Redmond, WA, United States), and a third reviewer (K.G.S) subsequently verified its accuracy.

   Previous studies have indicated that learning outcomes encompass motor learning, cognitive learning, affective learning, and social learning [3, 9]. Motor learning is defined as body growth-related physical characteristics and technical skills [3]; cognitive learning primarily

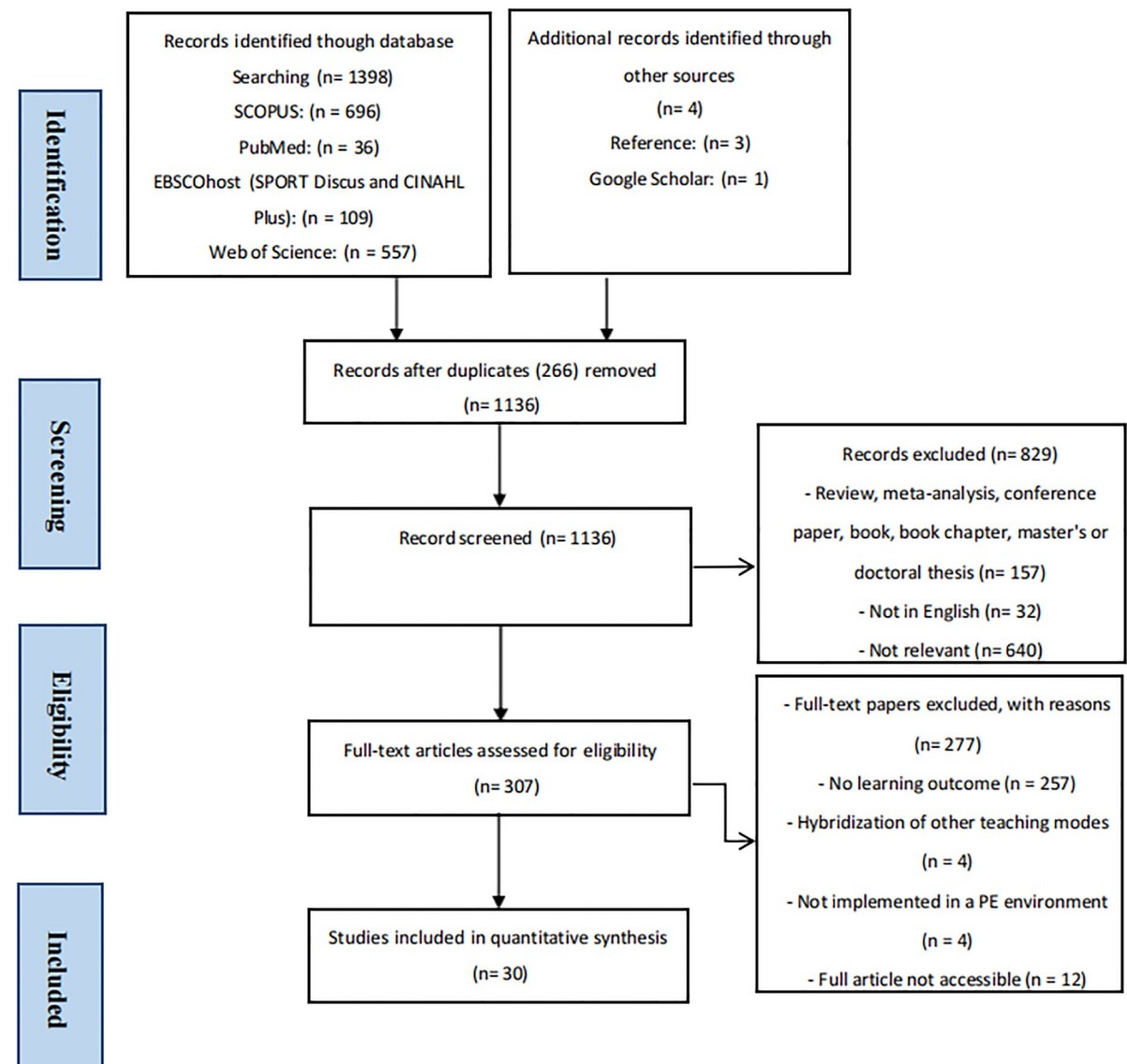

**Fig 1. PRISMA flow diagram.**

explains learning about strategies and decision-making abilities [63]; affective learning typically includes psychological factors such as confidence, self-esteem, motivation, and self-worth [33]; social learning comprises (a) interpersonal communication skills; (b) the ability to build relationships and listen to team members' opinions; (c) the sharing of beliefs, perspectives, and the collaborative construction of new understandings [10]. This serves as the reference basis for data extraction in this section.

## 2.5 Quality assessment and level of evidence

The updated PRISMA framework [36] was initially employed to assess the quality of this systematic review. Subsequently, the quality of published manuscripts was evaluated using a checklist adapted from the Strengthening the Reporting of Observational Studies in

**Table 1. Data extraction included in the study.**

| Sources | Purpose | Sample Characteristics | Hybrid | Length of Unit /Content | Data Collection | Study Design & Analysis | Learning Outcomes |
|---|---|---|---|---|---|---|---|
| Araújo et al. (2016) [37] Portugal | Analyzing students' improvements in game performance during a hybrid SEM-SGA approach | 17 grade 7 students, consisting of 7 girls and 10 boys, with an average age of 11.8 years | SEM—SGA | 25 lessons volleyball | Video Observation, Game Performance Assessment Tool (GPAI) | Quantitative: analysis of variance (ANOVA) | Motor, cognitive |
| Araújo et al. (2017) [38] Portugal | Examining the impact on student coaches' pedagogical content knowledge (PCK) | 21 students (11 males and 10 females); mean age 12.0 | SEM—SGA | 20–25 lessons volleyball | Video observation, field notes, interviews | Qualitative: thematic analysis method | Cognitive |
| Araújo et al. (2019) [39] Portugal | Analysis of student competition performance in three SEM-SGA seasons | 18 grade 7 students (8 female and 10 male) 11–13 years old | SEM—SGA | 20–25 lessons volleyball | Video observation, the Game Performance Assessment Instrument (GPAI) | Quantitative: hierarchical linear model | Motor |
| Araújo et al. (2020) [40] Portugal | Investigating students' tactical understanding | 96 students (40 boys and 53 girls, aged from 16–18 years-old) | SEM—SGA | 24 lessons volleyball | Video-based test | Quantitative: Mann-Whitney test | Motor, cognitive |
| Arikan (2020) [41] Turkey | Analysis of SEM-SEL on students' emotional intelligence levels | 166 students, aged from 15–17 years-old) | SEM—SEL | 16 lessons volleyball | The Schutte Emotional Intelligence Scale (SEI-S) | Quantitative: Covariance Analysis | Social |
| Buendía et al. (2021) [42] Spain | To compare the sportsmanship and enjoyment between the SEM-TGFU and PSRM | 85 teenagers, 39 girls, and 46 boys ages 16.42 ± 0.50 | SEM—TGFU | 10 lessons ultimate frisbee | The SSI questionnaires adapted to Physical Education (SSI-PE); the Multidimensional Sportsmanship Orientations Scale (MSOS) | Quantitative: Wilcoxon's test | Affective |
| Evangelio et al. (2021) [29] Spain | Explore students' perceptions of the SEM-CL-HBPE three-model mix | 115 grade 5–6 students (46.09% girls); 10–13 years old | SEM—CL-HBPE | 13 lessons an educative version of CrossFit | Interviews | Qualitative: thematic analysis method | Social |
| Farias et al. (2015) [43] Portugal | Analyzing the impact of SEM-IGCM on students' performance and understanding of soccer games | 24 grade 5 students, comprising 16 girls and 8 boys, with a mean age of 10.3 years | SEM—IGCM | 17 lessons soccer | The Game Performance Observation Instrument, Coding Association 6 Conference, The Game Understanding Test | Mixed studies: Mann-Whitney test, Wilcoxon test | Motor, cognitive |
| Farias et al. (2019) [44] Portugal | To examine game performance according to the tactical structures of invasion games throughout three consecutive model-based units | 26 students (10 females and 16 males); mean age of 12 years and three months | SEM—IGCM | Basketball: 20 lessons; Handball: 12 lessons; Football: 16 lessons | Game Performance Assessment Instrument (GPAI), Video observation | Quantitative: analyses of variance | Motor |
| Farias et al. (2022) [45] Portugal | To explore the influence of SEM-SGA on pre-service teachers' professional content knowledge (SCK) and students' play development | 60 eighth-grade students (three classes: 38 boys, 22 girls, Mage = 13.3) | SEM—SGA | 10 lessons volleyball | Using audio/ video data | Quantitative: analysis of variance; Mauchly's test; Bonferroni post hoc comparisons | Motor |
| Fernandez-Rio & Menendez (2017) [32] Spain | Evaluating the perceptions of students and teachers in an educational kickboxing learning unit | 71 grade 9 students, with an average age of 15.4 years and a standard deviation of 0.73 | SEM—TPSR | 16 lessons kickboxing | Open-ended questions, Photovoice, teacher and external observers' diaries, semi-structured interviews | Qualitative: thematic analysis method | Social |

(*Continued*)

**Table 1.** (Continued)

| Sources | Purpose | Sample Characteristics | Hybrid | Length of Unit /Content | Data Collection | Study Design & Analysis | Learning Outcomes |
|---|---|---|---|---|---|---|---|
| García-González et al. (2020) [4] Spain | Analyzing the effects of SEM-TGFU on student initial motivations | 49 students (M = 15.50, SD = 0.57) | SEM—TGFU | 10 lessons volleyball | Questionnaires on basic psychological need (BPN) support and satisfaction, novelty and variety satisfaction, motivation, and intention to be physically active | Quantitative: analysis of variance | Affective |
| Gil-Arias et al. (2017) [46] Spain | Assessing students' motivation levels for participating in physical activity | 55 grade 9 students, comprising 27 females and 28 males, with a mean age of 15.45 years | SEM—TGFU | 16 lessons volleyball | Scales: Autonomous motivation, Basic psychological needs, Enjoyment, Intention to be physically active | Quantitative: control group, MANOVA, Shapiro-Wilks test | Affective |
| Gil-Arias et al. (2020) [47] Spain | Analysis of SEM-TGFU on autonomy support, sensory Knowing the effects of motivating atmosphere, fun, and perceptual ability | 53 grade 9 students (16 female, 37 male); mean age 15.50 | SEM—TGFU | 16 lessons; handball and basketball | Physical Education Class Learning and Performance Orientation Questionnaire, Autonomy Support Coaching Strategies Questionnaire, Enjoyment and Perceived Ability Scale | Quantitative: a counter-balanced crossover design | Affective |
| Gil-Arias, Diloy-Peña, et al. (2020) [48] Spain | Analyzing the effects of SEM-TGFU on student motivational outcomes | 53 grade 9 students (16 female, 37 male); mean age 15.50 | SEM—TGFU | 10 lessons volleyball | Questionnaires, focus groups | Mixed: one-way analysis of variance, analysis of variance, deductive content analysis | Affective |
| Gil-Arias et al. (2021) [49] Spain | Investigating the effects of using SEM-TGFU on perceived autonomy support, perceived need satisfaction, autonomy motivation, and adaptive outcomes | 292 grade 6 students (140 female, 152 male); mean age 10.41 | SEM—TGFU | 16 lessons basketball | Autonomy Support Coaching Strategies Questionnaire, BPNs in Sport Scale, Perceived Causality Questionnaire, Relational Goals Questionnaire, Physical Activity Class Satisfaction Questionnaire | Quantitative: analysis of variance | Affective |
| Gouveia et al. (2022) [50] Portugal | Investigating the Impact of Different Pedagogical Models on Moderate-to-Vigorous Physical Activity in Physical Education Classes | 17 students, 9 males, 13.7 ± 0.90 years old | SEM—TGFU | 26 lessons | Objectively measure | Quantitative: Wilcoxon signed-rank test; analysis of variance | Motor |
| Guijarro & MacPhail. (2021) [51] Spain | Exploring Game Performance and Game Involvement through SEM-TGFU | 85 fourth- and fifth-grade students (aged 9–11) | SEM—TGFU | 15 lessons basketball | Systematic observation of video recordings of students' game behavior | Quantitative: multivariate analysis of variance | Motor |
| Hastie & Buchanan (2000) [52] The United States | To assess the effectiveness of SEM-TPSR in practice and formulate a theoretical model of Empowering Sports | United States, there are 45 boys in the sixth grade, aged 11 to 13 years old | SEM—TPSR | 26 lessons Xball | Independent observations, daily debriefs, and student interviews | Qualitative: constant comparison technique | Social |

(*Continued*)

**Table 1.** (*Continued*)

| Sources | Purpose | Sample Characteristics | Hybrid | Length of Unit /Content | Data Collection | Study Design & Analysis | Learning Outcomes |
|---|---|---|---|---|---|---|---|
| Hastie & Curtner-Smith (2006) [53] Australia | Analyzing the effects of implementing SEM-TGFU on both teachers and students | Australia; 29 grade 6 students, consisting of 11 boys and 18 girls, aged 11–12 years old | SEM—TGFU | 22 lessons batting/fielding games | Reflective logs, critical incident reflective sheets, tactical quizzes, game design forms, and team interviews | Analytic induction technique, enumerative analysis, typological analysis, and constant comparison | Cognitive, affective |
| Jia (2021) [54] China | Investigating the effects of SEM-TGFU on students' football cognitive performance and motor skills | 224 students | SEM—TGFU | 16-week (3 hours per week for a total of 48 hours) experimental | The ball games cognitive performance measurement (Prat et al., 2020); the benchmark of motor skills (Wang, 2018) | Quantitative: Analysis of variance | Cognitive; Motor |
| López-Lemus et al. (2023) [55] Spain | Analyzing the impact of SEM-TGFU on students' enjoyment, perceived competence, intention to be physically active, skill execution, decision-making, performance, and game involvement | CG: 70 students; age = 14.43 ± 0.693; n = 32 female) EG: (SEM-TGFU): 67 students; age = 13.91 ± 0.900; n = 30 female) | SEM—TGFU | 12 lessons handball | Game Performance Assessment Instrument (GPAI); the Enjoyment and perceived competence scale (ECS); the Measure of Intentionality to be Physically Active (MIFA) | Quantitative: pre-test/post-test quasi-experimental design | Cognitive; affective; Motor |
| Menendez & Fernandez-Rio (2017) [34] Spain | Analyzing the impact of SEM-TPSR on learners with disabilities during a contactless kickboxing learning unit | 12 students, with 5 of them having disabilities, and their ages range from 15 to 16 years | SEM—TPSR | 16 lessons kickboxing | Drawings, open-ended questions, discussion groups, diaries and semi-structured interviews | Qualitative: thematic analysis method | Social; Affective |
| Mesquita et al. (2012) [56] Portugal | Analyze the influence of SEM-IGCM on student decision-making, skill execution, and overall performance in competition | 26 grade 5 students, consisting of 17 girls and 9 boys, aged between 10 and 12 years old | SEM—IGCM | 22 lessons soccer | Game Performance Assessment Instrument (GPAI), Video observation | Quantitative: Mann-Whitney test, Wilcoxon test | Motor, cognitive |
| Oliveros & Fernandez-Rio (2022) [57] Spain | Investigating whether a hybrid pedagogical model could make a difference in adolescent girls' in-class physical activity levels | 66 students aged 13–17 years | SEM—TGFU | 12 lessons | Objectively measure | Quantitative: analysis of variance | Motor |
| Pan et al. (2023) [58] China | Comparing the learning effects between SEM-TGFU and TGFU on students' motivation, enjoyment, responsibility, and game performance | 90 students in the 4 classes (experimental group: 24 boys and 22 girls, Mage = 15.02 ± 0.73 years, and control group: 23 boys, and 21 girls, Mage = 14.78 ± 0.66 years) | SEM—TGFU | 10 weeks with 20 PE lessons | Responsibility scale in physical education (RSPE); Learning motivation scale in physical education (LMSPE); Sport enjoyment scale in physical education (SESPE); Game performance assessment instrument (GPAI) | Quantitative: A quasi-experimental design; Analysis of covariance | Cognitive; affective; Motor |
| Quiñonero-Martínez et al. (2023) [59] Spain | investigating the effect of the SEM-SPRM on students' physical fitness and physical activity | 76 Secondary Education students aged 12–14 (male: 32; female: 44) | SEM—SPRM | 17 lessons Colpbol sport; racket games; Frisbee Ultimate | Objectively measure | Quantitative: A quasi-experimental pre-post study | Motor |

(*Continued*)

**Table 1.** (Continued)

| Sources | Purpose | Sample Characteristics | Hybrid | Length of Unit /Content | Data Collection | Study Design & Analysis | Learning Outcomes |
|---|---|---|---|---|---|---|---|
| Silva et al. (2022) [60] Portugal | Investigating students' perceptions about lived learning experiences and active involvement in SEM-SGA | 25 students as participants (aged be- tween16 and17 years old) who | SEM— SGA | 26 lessons volleyball | Diary and interviews | Qualitative: thematic analysis method | Cognitive; effective |
| Stran et al. (2012) [61] The United States | To assess pre-service teachers' perceptions of SEM-TGFU and analyze the facilitators and barriers they encountered during model implementation | 22 pre-service teachers, comprising 14 males and 9 females, with an average age of 23. Additionally, there are 162 grade 5 students, aged 10–11 years old | SEM— TGFU | 20 lessons Invasion games | Focus group interviews, critical incident reflections, lesson plans, and observations | Qualitative: thematic analysis method | Cognitive, affective |
| Wei et al. (2020) [62] China | Analysis of the effects of SEM-TPSR on students' responsibility and exercise self-efficacy | 204 students | SEM— TPSR | 15-week (45 hours) | Questionnaires | Quantitative: regression analysis | Affective cognitive |

Epidemiology (STROBE) statement [64]. Referring to the 9 assessment criteria chosen by Shen and Shao for the structure of typical publications in this research domain: (1) Description of PM Hybridizations, (2) Characteristics of the Participants, (3) Reasonable Design of the Study, (4) Detailed Data Collection, (5) Detailed Data Analysis, (6) Validity and Reliability, (7) Inclusion of Models' Fidelity, (8) Report of Learning Outcomes, and (9) Discussion of Results. Each item was scored as 1 (yes) or 0 (no). The total quality score for each included study was determined by summing individual scores. Studies scoring 7 or above were categorized as "high quality," those scoring between 4 and 6 were classified as "moderate quality," and those scoring below 4 were classified as "low quality" [3]. Manuscripts had to score at least 4 points for inclusion. Two independent researchers (J.Z. and M.A.) assessed the selected studies. The final scores were reviewed and discussed by a research team composed of other co-authors, with any discrepancies negotiated with a third researcher (K.G.S) until a consensus was reached.

## 3. Results

### 3.1 Study background

The majority of the 30 studies were conducted in Western countries, with the highest number in Spain (13), followed by Portugal (10), China (3), the United States (2), Turkey (1), and Australia (1). The earliest article in this field was published in 2000 [52], with 20 articles published in the last five years, indicating that research on the hybridization of PM with SEM in physical education has become a recent focal point.

### 3.2 Participants

The majority of studies examined the effects of hybrid PMs in physical education classes for primary and secondary school students, with a combined sample size of 2,399 participants. Specifically, 70% of the studies focused on middle school students, totaling 1,238 participants, while seven studies targeted elementary school students, with 733 participants. Interestingly, no studies addressed university students. Additionally, two studies reported participant numbers but did not specify the ages of the students. The hybrid PMs curriculum primarily centered around sports such as soccer, basketball, volleyball, handball, taekwondo, frisbee, and various game-based activities like invasion games and batting/fielding games.

## 3.3 Hybrid curriculum implementation

Seven hybrid types were identified across all literature. One study employed a combination of three PMs: SEM-CL-HBPE [29], while the rest involved combinations of two PMs: SEM-TGFU (14), SEM-SGA (6), SEM-TPSR (4), SEM-IGCM (3), SEM-SEL (1), SEM-SPRM (1). Most studies emphasized the organization structure based on SEM seasons, incorporating elements of other PMs, forming a "SEM + 1" model. The learning tasks and teaching content of SEM seasons mainly originated from game-centered models (TGFU, IGCM, and SGA) and TPSR.

## 3.4 Study design and data collection

Different research designs were employed for various instructional objectives. Qualitative studies primarily collected data through interviews and observations, while quantitative studies mainly utilized scales and questionnaires. There was a total of 20 quantitative studies, 7 qualitative studies, and 3 mixed-methods studies.

## 3.5 Quality assessment and level of evidence

The results of the quality assessment are presented in Table 2, indicating that all 30 studies are of high quality. The primary factors influencing article quality are validity and reliability, and the inclusion of models' fidelity. Among these, 12 studies did not conduct validity and reliability testing, and 22 studies lacked models' fidelity procedures.

## 3.6 Optimization of learning outcomes by hybrid pedagogical model

This section aims to demonstrate the impact of hybrid PMs included in the literature on learning outcomes.

**3.6.1 The impact of hybrid SEM-TGFU on students' learning outcomes.** Existing research indicates that hybrid SEM-TGFU benefits students in terms of motor learning, cognitive learning, and affective learning outcomes.

Motor Learning: Stran et al. found that SEM-TGFU increased student engagement in the classroom [61]. Guijarro & MacPhail concluded that the use of hybrid SEM-TGFU units surpassed SEM in decision-making, support, overall game performance, and game participation, leading to improved game performance and engagement [51]. Jia found a significant impact of SEM-TGFU on students' motor skills [54]. Notably, Oliveros & Fernandez-Rio's results showed that female students scored significantly lower than males in high-intensity physical activity (MVPA) during hybrid SEM-TGFU, emphasizing that the hybrid PM itself may not assist girls in achieving MVPA scores similar to boys. Additionally, the study found that MVPA for lower-grade students was significantly higher than for higher-grade students [57]. Gouveia et al. concluded that SEM-TGFU significantly reduced classroom sedentary time [50]. López-Lemus et al. [55] demonstrated that the implementation of hybrid models SEM/TGFU could enhance students' game involvement and performance, as well as increase enjoyment, perceived competence, and intention to be physically active, in both boys and girls.

Cognitive Learning: Hastie & Curtner-Smith found that students could understand, appreciate, and execute some basic hitting, throwing, and outfield throwing tactics and strategies, as well as grasp the overall principles, rules, and structures of hitting/batting/fielding games, realizing their importance [53]. Jia found a significant impact of SEM-TGFU on students' cognitive performance [54].

Affective Learning: Gil-Arias et al. reported that students showed significant improvements in autonomy, competence, and enjoyment [46]. Gil-Arias et al. further found a significant

**Table 2. Study quality checklist with quality scores assigned.**

| Author(s)/Date | Description of PM Hybridizations | Characteristics of the Participants | Reasonable Design of the Study | Detailed Data Collection | Detailed Data Analysis | Validity and Reliability | Inclusion of Models' Fidelity | Report of Learning Outcomes | Discussion of Results | quality score | level of evidence |
|---|---|---|---|---|---|---|---|---|---|---|---|
| Araújo et al. (2016) [37] | 1 | 1 | 1 | 1 | 1 | 1 | 1 | 1 | 1 | 9 | High |
| Araújo et al. (2017) [38] | 1 | 1 | 1 | 1 | 1 | 0 | 0 | 1 | 1 | 7 | High |
| Araújo et al. (2019) [39] | 1 | 1 | 1 | 1 | 1 | 0 | 1 | 1 | 1 | 8 | High |
| Araújo et al. (2020) [40] | 1 | 1 | 1 | 1 | 1 | 1 | 0 | 1 | 1 | 8 | High |
| Arikan (2020) [41] | 1 | 1 | 1 | 1 | 1 | 1 | 0 | 1 | 1 | 8 | High |
| Buendía et al. (2021) [42] | 1 | 1 | 1 | 1 | 1 | 1 | 0 | 1 | 1 | 8 | High |
| Evangelio et al. (2021) [29] | 1 | 1 | 1 | 1 | 1 | 1 | 0 | 1 | 1 | 8 | High |
| Farias et al. (2015) [43] | 1 | 1 | 1 | 1 | 1 | 1 | 0 | 1 | 1 | 8 | High |
| Farias et al. (2019) [44] | 1 | 1 | 1 | 1 | 1 | 0 | 1 | 1 | 1 | 8 | High |
| Farias et al. (2022) [45] | 1 | 1 | 1 | 1 | 1 | 1 | 1 | 1 | 1 | 9 | High |
| Fernandez-Rio & Menendez (2017) [32] | 1 | 1 | 1 | 1 | 1 | 0 | 0 | 1 | 1 | 7 | High |
| García-González et al. (2020) [4] | 1 | 1 | 1 | 1 | 1 | 1 | 0 | 1 | 1 | 8 | High |
| Gil-Arias et al. (2017) [46] | 1 | 1 | 1 | 1 | 1 | 1 | 0 | 1 | 1 | 8 | High |
| Gil-Arias et al. (2020) [47] | 1 | 1 | 1 | 1 | 1 | 1 | 0 | 1 | 1 | 8 | High |
| Gil-Arias, Diloy-Peña, et al. (2020) [48] | 1 | 1 | 1 | 1 | 1 | 1 | 0 | 1 | 1 | 8 | High |
| Gil-Arias et al. (2021) [49] | 1 | 1 | 1 | 1 | 1 | 1 | 1 | 1 | 1 | 9 | High |
| Gouveia et al. (2022) [50] | 1 | 1 | 1 | 1 | 1 | 0 | 0 | 1 | 1 | 7 | High |
| Guijarro & MacPhail. (2021) [51] | 1 | 1 | 1 | 1 | 1 | 1 | 0 | 1 | 1 | 8 | High |
| Hastie & Buchanan (2000) [52] | 1 | 1 | 1 | 1 | 1 | 1 | 0 | 1 | 1 | 8 | High |
| Hastie & Curtner-Smith (2006) [53] | 1 | 1 | 1 | 1 | 1 | 0 | 0 | 1 | 1 | 7 | High |
| Jia (2021) [54] | 1 | 1 | 1 | 1 | 1 | 0 | 0 | 1 | 1 | 7 | High |
| López-Lemus et al. (2023) [55] | 1 | 1 | 1 | 1 | 1 | 1 | 1 | 1 | 1 | 9 | High |

*(Continued)*

**Table 2.** (Continued)

| Author(s)/Date | Description of PM Hybridizations | Characteristics of the Participants | Reasonable Design of the Study | Detailed Data Collection | Detailed Data Analysis | Validity and Reliability | Inclusion of Models' Fidelity | Report of Learning Outcomes | Discussion of Results | quality score | level of evidence |
|---|---|---|---|---|---|---|---|---|---|---|---|
| Menendez & Fernandez-Rio (2017) [34] | 1 | 1 | 1 | 1 | 1 | 0 | 0 | 1 | 1 | 7 | High |
| Mesquita et al. (2012) [56] | 1 | 1 | 1 | 1 | 1 | 1 | 0 | 1 | 1 | 8 | High |
| Oliveros & Fernandez-Rio (2022) [57] | 1 | 1 | 1 | 1 | 1 | 1 | 0 | 1 | 1 | 8 | High |
| Pan et al. (2023) [58] | 1 | 1 | 1 | 1 | 1 | 1 | 1 | 1 | 1 | 9 | High |
| Quiñonero-Martínez et al. (2023) [59] | 1 | 1 | 1 | 1 | 1 | 0 | 0 | 1 | 1 | 7 | High |
| Silva et al. (2022) [60] | 1 | 1 | 1 | 1 | 1 | 0 | 0 | 1 | 1 | 7 | High |
| Stran et al. (2012) [61] | 1 | 1 | 1 | 1 | 1 | 0 | 1 | 1 | 1 | 8 | High |
| Wei et al. (2020) [62] | 1 | 1 | 1 | 1 | 1 | 0 | 0 | 1 | 1 | 7 | High |

increase in students' input, praise for autonomous behavior, perceived competence, and enjoyment [47]. Two additional articles from their team, published in the same year, added comparisons between students of different genders, highlighting a larger effect size in girls, emphasizing the importance of TGFU/SEM units in improving student motivation outcomes, especially for girls [51, 55]. Additionally, García-González et al. studied the impact of hybrid SEM-TGFU on students at different Relative Autonomy Index (RAI) levels, emphasizing a larger effect size in "medium" or "low" level students, although SEM-TGFU units in volleyball teaching were beneficial to students at all three levels [4]. The results of Buendía et al.'s study showed a significant difference in improving student enjoyment with SEM-TGFU [42]. Pan et al. found that SEM-TGFU had positive learning effects on students' motivation, enjoyment of physical activity, sense of responsibility, and game performance, surpassing the effects of the TGFU model [58].

**3.6.2 The impact of hybrid SEM-SGA on students' learning outcomes.** Existing research indicates that hybrid SEM-TGFU benefits students in terms of motor learning, cognitive learning, affective learning, and social learning outcomes.

Motor Learning: Araújo et al. conducted three assessments of game performance using hybrid SEM-SGA for students of different genders and skill levels. The results showed improvement in all indicators of game performance, including game performance, game involvement, decision making, adjustment, skill efficiency, and skill efficacy, for both boys and girls from pre-test to post-test. Additionally, students with lower skill levels gained greater benefits, and there were no significant changes in all indicators between the post-test and retention test [37]. This reinforces the idea that the implementation of this hybrid approach in the future should be adjusted based on different skill levels in terms of content and learning tasks. Araújo et al. applied hybrid SEM-SGA to study the long-term development of students' volleyball play performance over three years. The results of the survey showed an improvement in the play performance levels of all 18 students, and the study also concluded that the implementation

over three seasons would gradually eliminate the gap in students' skill levels [39]. It implies that the improvement effect of students with low skill levels is higher than that of students with high skill levels. Farias et al. investigated students' game-play development using hybrid SEM-SGA, revealing improvements in most game variables (such as "serving," "receiving," and "setting") and the "game performance index" [45].

Cognitive Learning: Araújo et al. studied students' pedagogical content knowledge of coaches using hybrid SEM-SGA, demonstrating the effectiveness of SEM-SGA in enhancing students' coaches' pedagogical content knowledge [38]. Araújo et al. proposed that students' tactical understanding during hybrid SEM-SGA instruction significantly improved [40]. Farias et al. investigated the specialized content knowledge (SCK) of pre-service teachers using hybrid SEM-SGA, revealing a significant improvement in pre-service teachers' SCK over time [45].

Affective Learning and Social Learning: Existing research suggests that hybrid SEM-SGA benefits students in terms of cognitive learning, affective learning, and social learning outcomes. Silva et al. conducted a unique survey analysis of pre-service teachers using hybrid SEM-SGA, employing inductive and deductive theme analysis. The results showed that the use of this hybrid approach helped teachers act as facilitators of learning, increased students' levels of responsibility for their learning experiences, promoted students' autonomy and sense of active control over classroom activities, facilitated the development of student's abilities and volleyball fundamental knowledge, and increased their interest and engagement in physical education [60].

**3.6.3 The impact of hybrid SEM-TPSR on students' learning outcomes.** Existing research indicates that hybrid SEM-TPSR benefits students in terms of motor learning and social learning outcomes. Hastie & Buchanan concluded that hybrid SEM-TPSR allows achievement in a powerful triad of goals-motor skill competence, social responsibility, and personal empowerment [52]. Menendez-Santurio & Fernandez-Rio conducted hybrid SEM-TPSR teaching targeting students with disabilities and found that it helped them and their peers establish connections both inside and outside the classroom [34]. This result seems to broaden the range of beneficiaries. At the same time, it enriches the future research direction. The results of Fernandez-Rio & Menendez-Santurio's study indicated that SEM-TPSR significantly enhances students' responsibility and exercise self-efficacy [32].

**3.6.4 The impact of hybrid SEM-IGCM on students' learning outcomes.** Existing research indicates that hybrid SEM-IGCM benefits students in terms of motor learning, cognitive learning, and social learning outcomes.

Motor Learning and Social Learning: Mesquita et al. provided students with opportunities to improve skill execution and tactical decision-making through a hybrid SEM-IGCM soccer unit. Furthermore, inclusive participation had a strong impact on learning for girls and students with lower skill levels [56]. Farias et al. concluded that hybrid SEM-IGCM promoted improvements in students' soccer game performance and understanding, increasing the correlation between the two concepts [43]. Farias et al. examined game performance in three consecutive basketball, handball, and soccer units based on the hybrid SEM-IGCM model. The results showed significant improvements in game performance for handball and soccer units, but not for basketball. This means that hybrid PMs may have different effects on different sports. However, when breaking down game performance into tactical structure indicators, improvements were observed in all seasons. The correlation between tactical structure indicators and game performance increased over time. The predictive model of game performance improvement demonstrated correlations with expanded team contextual features, coordinated interpersonal dynamics in the game, the nature of peer teaching mediation, and the game format [44].

**3.6.5 The impact of hybrid SEM-SEL, SEM-SPRM, and SEM-CL-HBPE on students' learning outcomes.**   Due to the limited number of included studies for hybrid SEM-SEL, SEM-SPRM, and SEM-CL-HBPE (each with only one study), they will be collectively described here. Evangelio et al.'s study is the only one among the included literature that hybridized all three models. The results indicate that the current hybrid SEM-CL-HBPE can contribute to shaping habitual, motivated, critical, and informed advocates [29].

Using a 16-week SEM-SEL volleyball program based on the SEM, Arikan (2020) compiled a SEL. The results showed that the SEM-SEL volleyball program is effective in enhancing students' emotional intelligence levels, advocating for further exploration of the compatibility of SEM and SEL hybridization.

Quiñonero-Martínez et al.'s results revealed different outcomes in motor learning. Traditional methodology, including standing long jump and speed-agility scores, showed significant improvement. Conversely, in SEM-SPRM, neither of the tests demonstrated significant improvement [59].

## 4. Discussion

### 4.1 Hybrid curriculum implementation

Our review of the literature reveals a lack of detailed descriptions regarding fixed or optimal methods for implementing the "SEM + 1" model. This may be due to the inherent flexibility of the "SEM + 1" approach, which allows teaching strategies to be adjusted based on specific instructional goals and contextual factors, effectively leveraging the strengths of both models [8]. Some studies have provided intervention programs that emphasize combining the characteristics and principles of the two models, offering guidance to both future researchers and frontline teachers [48, 58]. However, certain challenges remain in implementing these hybrid models. First, these intervention programs are often developed in specific educational contexts, which may lead to issues with adaptability when applied in different settings. Second, while some studies offer guidance, they may not fully address the practical difficulties teachers face and the professional support required for effective implementation, potentially impacting the model's effectiveness. Additionally, there is often a gap between the theoretical design of these hybrid models and their practical execution in real-world classrooms. Teachers may struggle with balancing the dual demands of integrating two pedagogical models while maintaining coherence in their teaching, which can hinder the overall implementation process. Future efforts should focus on providing clearer implementation guidelines and professional development opportunities to better support educators in adopting these hybrid approaches.

### 4.2 The impact of hybrid SEM-TGFU on students' learning outcomes

The popularity of SEM-TGfU as a hybrid pedagogical model can be attributed to three key factors: the theoretical foundation for constructing these PMs [32, 63, 64], the characteristics of the models themselves [9], and the motivational aspects they bring to physical education [65].

Firstly, while constructivist learning theory and game theory serve as the theoretical underpinnings for both SEM and TGfU, it is important to highlight that these theories support the effectiveness of the hybrid approach. Constructivism provides the rationale for creating active, student-centered learning environments, which is the cornerstone of SEM and TGfU. This theoretical foundation justifies why combining these models can enhance students' learning experiences by promoting autonomy and engagement [32, 63, 64].

Secondly, the distinct characteristics of SEM and TGfU complement each other, making their integration beneficial for students. SEM's emphasis on structured seasons, affiliation, formal competition, culminating events, record-keeping, and festivity helps students experience

authentic sporting events through various roles [9]. However, SEM may not sufficiently address tactical understanding and skill development [65]. TGfU, on the other hand, fills this gap by focusing on game-based learning, tactical awareness, and skill execution through a structured approach [66]. By integrating TGfU's instructional steps into SEM's framework, students gain a more comprehensive understanding of sports, which includes both the organizational structure and tactical execution [46].

Lastly, the motivational aspects of the hybrid SEM-TGfU model further enhance student engagement. Both models encourage cooperative learning, which fosters a positive team environment and supports improved decision-making, game performance, and overall involvement [67]. This approach also reduces sedentary behavior, increases physical activity, and positively influences students' motivation, enjoyment, and sense of responsibility [45, 68].

## 4.3 The impact of hybrid SEM-SGA on students' learning outcomes

In addition to research on hybrid SEM-TGFU, there is a relatively significant amount of research on SEM-SGA. SAG emphasizes providing students with an appropriate framework for the development of tactical and technical skills in net sports to ensure the success of the game [37, 56].

Combining SEM with SGA retains SEM's core elements such as stable teams, formal competition, and role-playing, while integrating SGA's learning tasks and sports skills taught during the season. This combination enables students to enhance the acquisition of motor skills and performance in the game [37, 39, 45]. The primary purpose of this hybrid is also to address SEM's lack of specific teaching strategies to cultivate students' tactical competence [37].

## 4.4 The impact of hybrid SEM-TPSR on students' learning outcomes

The integration of TPSR with SEM leverages SEM's six characteristics (seasons, affiliation, formal competition, culminating event, record keeping, and festivity) to provide a structured and authentic learning environment. SEM-TPSR encourages cooperation, self-esteem development, and sustained social interactions [34]. This hybrid model supports students' autonomy and responsibility, fostering attention to their rights, feelings, and the needs of others [32, 34].

In investigating how SEM functions as a foundational model, it becomes clear that the organizational strengths of SEM complement TPSR's focus on personal and social responsibility. This synergy addresses key factors supporting their integration, such as shared theoretical foundations and mutual reinforcement of learning outcomes. Consequently, SEM-TPSR optimizes learning outcomes by promoting interpersonal skills, social responsibility, and overall student motivation [10, 33, 69].

## 4.5 The impact of hybrid SEM-IGCM on students' learning outcomes

SEM-IGCM is also a prevalent choice in hybrid teaching due to its alignment with constructivist principles and the complementary strengths of both models. The Invasion Game Competence Model (IGCM), like the Step-Game Approach (SGA) and Teaching Personal and Social Responsibility (TPSR), originates from Teaching Games for Understanding (TGfU) and is rooted in constructivist learning theory and game theory. IGCM focuses on game-centered approaches, emphasizing structured planning, the development of tactical skills, decision-making, and skill execution in invasion sports. It provides a comprehensive framework for assessing competence in invasion games, considering overall game performance and participation.

When combined with the Sport Education Model (SEM), IGCM benefits from SEM's structured framework, including stable teams, formal competitions, and role-playing, while IGCM

enhances the learning plans and sports skills taught during the season. This hybrid approach allows students to improve skill acquisition and game performance [39, 43, 56].

## 4.6 The impact of hybrid SEM-SEL, SEM-SPRM, and SEM-CL-HBPE on students' learning outcomes

The hybrid models of SEM-SEL, SEM-SPRM, and SEM-CL-HBPE facilitate the development of students' sociological and affective learning abilities [29, 40]. The emphasis on cooperative learning and student engagement in these models likely contributes to their positive impact. SEM's stable teams create a consistent platform for fostering social skills through collaborative learning [70], while HBPE promotes healthy lifestyles, motivating team members to prioritize physical well-being [71]. This group teaching approach enhances students' ability to work within teams, improving social relationships. However, further research is needed due to the limited evidence in the literature.

Additionally, in SEM-SPRM, performance tests for standing long jump and speed and agility did not show significant improvement, whereas the traditional teaching group demonstrated significant enhancement. This suggests a need for further clarification on the compatibility of SEM and SEL hybrids [3].

## 4.7 Implementation limitations and future applications

First of all, since the premise of hybrid PM is that teachers have a comprehensive understanding of the theories, methods, and procedures of each mode and have teaching experience in implementing them, this creates major obstacles and challenges for pre-service teachers [34, 48, 61]. At the same time, teachers need to accurately select suitable hybrid pedagogical models (PMs) based on teaching needs. The implementation of hybrid PMs often centers around the six key features of the Sport Education Model (SEM). Teachers must repeatedly emphasize basic team agreements, individual roles, and how to conduct independent team activities. This focus can sometimes reduce the time available for the instructional content of the other teaching model, which may impact the overall progress of the class [53]. Therefore, in the future application of hybrid PMs, it is important to strengthen the early training of PE teachers on the use of hybridization. This training should include a focus on maintaining model fidelity, ensuring that teachers can accurately and consistently implement the hybrid models as intended. Without this fidelity, the benefits of hybrid PMs may not be fully realized, and the variability in implementation could lead to inconsistent outcomes. PE teachers should accurately understand student needs, consider the most suitable hybrid PM, and master the two PMs in hybridization. If implementation challenges arise or teaching progress is hindered, selecting one model may be advisable. Moreover, proficient teachers should adjust teaching content and tasks according to students' skill levels to optimize learning outcomes.

## 5. Conclusion

This review examines hybrid pedagogical models (PMs) integrated with SEM to optimize students' learning outcomes. It identifies seven hybrid PMs, primarily composed of two models, with the blend of SEM and Teaching Games for Understanding (TGfU) emerging as the mainstream in current research. SEM serves as the foundational structure, creating a stable framework for these hybrids, referred to as "SEM + 1 models." These hybrids have shown positive effects on enhancing students' learning outcomes across various domains. While the motivational aspects of these models contribute to improved learning outcomes, evidence for some hybrid models remains limited. Future research should focus on exploring diverse interventions and addressing issues related to teaching coherence and teacher competence. Ensuring

model fidelity and expanding empirical support are essential for the effective development and implementation of these hybrid models.

## Supporting information

**S1 Table. Detailed search strategy.**
(DOC)

**S1 Checklist. PRISMA 2020 checklist.**
(PDF)

**S1 File. Numbering table.**
(XLS)

## Author Contributions

**Conceptualization:** Junlong Zhang, Kim Geok Soh, Mohd Ashraff Mohd Anuar.

**Investigation:** Junlong Zhang, Xiaorong Bai.

**Methodology:** Kim Geok Soh, Mohd Ashraff Mohd Anuar.

**Supervision:** Kim Geok Soh.

**Writing – original draft:** Junlong Zhang.

**Writing – review & editing:** Kim Geok Soh, Wensheng Xiao.

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
