## [Decision Letter · Decision Letter 0]

15 Aug 2024

PONE-D-24-20221Optimizing learning outcomes in Physical Education: A comprehensive systematic review of hybrid pedagogical models integrated with the Sport Education ModelPLOS ONE

Dear Dr. zhang,

Thank you for submitting your manuscript to PLOS ONE. After careful consideration, we feel that it has merit but does not fully meet PLOS ONE’s publication criteria as it currently stands. Therefore, we invite you to submit a revised version of the manuscript that addresses the points raised during the review process.

 Please submit your revised manuscript by Sep 29 2024 11:59PM. If you will need more time than this to complete your revisions, please reply to this message or contact the journal office at plosone@plos.org. Please include the following items when submitting your revised manuscript:A rebuttal letter that responds to each point raised by the academic editor and reviewer(s). You should upload this letter as a separate file labeled 'Response to Reviewers'.A marked-up copy of your manuscript that highlights changes made to the original version. You should upload this as a separate file labeled 'Revised Manuscript with Track Changes'.An unmarked version of your revised paper without tracked changes. You should upload this as a separate file labeled 'Manuscript'.

We look forward to receiving your revised manuscript.

Kind regards,

Musa Adekunle Ayanwale

Academic Editor

PLOS ONE

Journal Requirements: 

2. We note that your Data Availability Statement is currently as follows: [All relevant data are within the manuscript and its Supporting Information files]

Reviewers comment

**Comments to the Author**

1. Is the manuscript technically sound, and do the data support the conclusions?

Reviewer #1: Partly

Reviewer #2: Yes

2. Has the statistical analysis been performed appropriately and rigorously? 

Reviewer #1: N/A

Reviewer #2: Yes

3. Have the authors made all data underlying the findings in their manuscript fully available?

Reviewer #1: Yes

Reviewer #2: Yes

4. Is the manuscript presented in an intelligible fashion and written in standard English?

Reviewer #1: Yes

Reviewer #2: Yes

5. Review Comments to the Author

Reviewer #1: General Comments for Authors

Dear authors, the manuscript is logically presented. The PRISMA steps included for the selection of studies is also well implemented and quality of study assigned. My primary concern relates to the potency of the findings in adding to the extant literature in this domain and specifically these concerns primarily relate to the practical application.

I have my doubts about the contribution of this review. The stipulated purpose of this study was “to conduct a comprehensive systematic review of the role of hybrid PMs, combined with the SEM, in optimizing learning outcomes in physical education. Specifically, this study will investigate the impact of these hybrid models, integrated with SEM, on students' learning outcomes. Additionally, the study seeks to explore the advantages of SEM as a foundational model in hybrid PMs”. However, significant work is required to get this review to the point where I would consider it publishable. Details of this issue and specific comments are detailed below in the hope that they provide the author(s) reflections on making the empirical analysis more rigorous and insightful.

Regarding format, please check all text for font size and typeface because there are inconsistencies.

Specific Comments

Introduction:

In the theoretical framework I miss a paragraph explaining the sport education model and how it has hybridised with other models. There is only one paragraph in which baggage is presented and another one in which some improvements produced by the model in some studies are shown.

On the other hand, I have detected errors in the quotations and references. Please, this should be revised. For example on Page 5, L125 the numbers of citations do not correspond with the papers.

Shen & Shao 130 further investigate the impact and mechanisms of hybrid models on students' learning outcomes. What is the difference between this work and Shen's review?

Method

Figure 1: The words in the blue squares in figure 1 have been cut off, I think the last letter is missing

The tables showing the studies are sorted by year (Tables 1 and 2). Please sort them alphabetically to better identify them. Besides, the citations should be numbered as in the citations. And even marked in the references. Otherwise, we cannot identify them and have not been able to check that they have all been mentioned in the results and discussion. Please check this aspect as well.

Results

The sentence of the line 226-227: “The total student sample size across these 30 articles reached 2381 individuals, with the majority of students aged between 10 and 16 years”. This sentence does not provide any useful data for authors, it is a summation and does not specify anything. Please do some ranking of the number of articles and ages as well as an average number of participants.

P15, Line 259: there are two separate square brackets attached to the text.

P16, L267: Lccópez-Lemus et al. this quote is not referenced. Please check throughout the text so that this does not happen with any of the following. I think you refer to López-Temus.

Discussion

P20-21, L373-383: Why you discuss that most articles are from TGfU and SEM because of constructivism and expose their characteristics in the discussion. And again in 392 you justify that also are built on the foundation of constructivist learning theory.

P22, L402-406: Why here again the theory of the model is explained. It is about discussing, not expounding theory that should be part of the theoretical framework. In this case of TPSR.

The article would contribute more if it showed how to implement these hybridisations.

4.5 Implementation limitations and future applications

Can they indicate why you think that L454 “resulting in the teaching content of the other mode of teaching is often interrupted”?

Please, review long sentences: example L450-454.

Conclusion

Please review the 1st sentence of the conclusion “This systematic review systematically examines”

In general, the conclusions do not respond to the aims of the review.

References

Please check the dois in all the references, in Shen's reference does not work.

Reviewer #2: The author(s) did a good review which shows that they understand the concept that is been review. I am recommending that the manuscript be accepted for publication. The author(s) should take second look at the the language and ensure grammatical errors are corrected.

6. PLOS authors have the option to publish the peer review history of their article (what does this mean?). If published, this will include your full peer review and any attached files.

Reviewer #1: No

Reviewer #2: **Yes**

---

## [Author Response · Author response to Decision Letter 0]

28 Aug 2024

Reviewer Comments and revisions

(Note: The text marked in different colors represents the reviewer's comments, while the black font indicates our revisions and responses.)

Reviewer #1:

General Comments for Authors: 

Dear authors, the manuscript is logically presented. The PRISMA steps included for the selection of studies is also well implemented and quality of study assigned. My primary concern relates to the potency of the findings in adding to the extant literature in this domain and specifically these concerns primarily relate to the practical application.

We want to sincerely thank you for your time and effort in reviewing our manuscript. Your thorough and rigorous comments, along with your insightful feedback, have significantly contributed to improving the quality of this work. We are truly grateful for your contributions toward the successful publication of our manuscript. The changes we have made in response to your comments are marked in blue and bold.

Regarding format, please check all text for font size and typeface because there are inconsistencies.

We are very sorry for our incorrect writing, and it has been rectified.

Specific Comments: 

Introduction

In the theoretical framework I miss a paragraph explaining the sport education model and how it has hybridised with other models. There is only one paragraph in which baggage is presented and another one in which some improvements produced by the model in some studies are shown.

We fully agree with your suggestion and have added a paragraph that provides examples from previous literature on how the Sport Education Model (SEM) has been hybridized with the Teaching Games for Understanding (TGfU) model. Additionally, we have ensured that the new content is well-integrated with the surrounding context for better coherence. (P3, Line 111-122)

On the other hand, I have detected errors in the quotations and references. Please, this should be revised. For example on Page 5, L125 the numbers of citations do not correspond with the papers.

We are very sorry for our errors in the quotations and references, and it has been rectified. (P4, Line 143)

Shen & Shao 130 further investigate the impact and mechanisms of hybrid models on students' learning outcomes. What is the difference between this work and Shen's review?

Thank you for your valuable feedback. We have identified several key differences between our study and the research by Shen & Shao:

Research Objectives: The primary distinction lies in the research objectives. Shen & Shao’s work builds upon the systematic review conducted by González-Víllora et al. (González-Víllora, S.; Evangelio, C.; Sierra-Díaz, J.; Fernandez-Rio, J. Hybridizing Pedagogical Models: A Systematic Review. Eur. Phys. Educ. Rev. 2019, 25, 1056–1074) and further examines the impact of various hybrid pedagogical models on student learning outcomes and their mechanisms. In contrast, our study advances this field of inquiry by focusing specifically on hybrid models involving the Sport Education Model (SEM). Our research is more targeted, concentrating on how different pedagogical models are hybridized with SEM, a model that has increasingly become the centerpiece of hybrid PM teaching. However, any further research is inherently built upon the contributions of previous scholars.

Methodological Differences: Significant methodological differences also exist between our study and that of Shen & Shao. Differences in search terms, databases, inclusion/exclusion criteria, and updated timeframes led to a larger dataset in our review, with 30 studies included compared to Shen & Shao’s 17. Notably, 10 of the additional studies in our review were published within the last three years. These recent contributions not only diversify our findings but also provide a more comprehensive and current view of hybrid pedagogical models, thereby enriching and potentially altering the outcomes.

Research Contributions:Research Contributions: Shen and Shao's contributions primarily focus on categorizing hybrid models based on their impact on student learning outcomes across four dimensions: motor, cognitive, affective, and social learning. They also analyze the underlying mechanisms of these hybridizations, concluding that the duration of hybrid model implementation and teachers' familiarity with these models are key limiting factors. However, our study highlights a different aspect: the "SEM+1" model has emerged as the dominant trend in current hybrid PM research. We conducted a targeted analysis of the factors that support the hybridization of SEM with other models and examined SEM's role as the foundational framework in these combinations. Furthermore, we explored how each "SEM+1" model specifically contributes to optimizing student learning outcomes.

Identified Issues: The issues identified in our study also differ from those highlighted by Shen & Shao. While Shen & Shao suggest future research should focus on quasi-experimental studies comparing hybrid models with single models to assess their relative advantages, our study raises more specific concerns. First, we identify a lack of evidence supporting some "SEM+1" models. Second, we emphasize the importance of maintaining fidelity to the "SEM+1" model in future research, ensuring classroom coherence, addressing teacher competency issues, and solving the issue of time allocation between the two models.

In summary, while Shen & Shao provide a broad categorization and analysis of hybrid models, our study takes a more focused approach, specifically analyzing SEM-based hybrid models, identifying current trends, and offering a detailed exploration of their educational implications.

Method

Figure 1: The words in the blue squares in figure 1 have been cut off, I think the last letter is missing

Thank you very much for your meticulous review. We sincerely apologize for the error in Figure 1 and have corrected it.

The tables showing the studies are sorted by year (Tables 1 and 2). Please sort them alphabetically to better identify them. Besides, the citations should be numbered as in the citations. And even marked in the references. Otherwise, we cannot identify them and have not been able to check that they have all been mentioned in the results and discussion. Please check this aspect as well.

Thank you for your valuable feedback. We have revised the manuscript as per your suggestions.

Results

The sentence of the line 226-227: “The total student sample size across these 30 articles reached 2381 individuals, with the majority of students aged between 10 and 16 years”. This sentence does not provide any useful data for authors, it is a summation and does not specify anything. Please do some ranking of the number of articles and ages as well as an average number of participants.

Thank you for your insightful suggestion. We agree that the sentence could be more informative and have revised it accordingly. We have now provided a breakdown of the number of articles by age groups, along with the average number of participants. This additional detail will give readers a clearer understanding of the distribution of the sample sizes and age ranges across the studies. (P9, line 241-248, 3.2 Participants)

P15, Line 259: there are two separate square brackets attached to the text.

P16, L267: Lccópez-Lemus et al. this quote is not referenced. Please check throughout the text so that this does not happen with any of the following. I think you refer to López-Temus.

Thank you very much for your meticulous review. We are very sorry for our incorrect writing, and it has been rectified.

Discussion

P20-21, L373-383: Why you discuss that most articles are from TGfU and SEM because of constructivism and expose their characteristics in the discussion. And again in 392 you justify that also are built on the foundation of constructivist learning theory.

Thank you for your insightful feedback. We appreciate your point about the focus of the discussion section. To clarify, we included constructivism as a theoretical factor in our discussion due to its established role in the literature on SEM and TGfU. For instance, Hastie and Curtner-Smith (2006) note that both SEM and TGfU are supported by constructivism, which emphasizes active learning, problem-solving, and decision-making [1]. This perspective is consistent with Dyson et al. (2004), who also highlight constructivism as a key factor in the effectiveness of these models in enhancing student learning outcomes [1]. 

Therefore, our discussion integrates constructivism to illustrate its relevance in explaining the effectiveness of the hybrid SEM-TGfU model. However, we acknowledge that there was room for improvement in this section and have made revisions accordingly. We have improved The content of "4.2 The impact of hybrid SEM-TGFU on students' learning outcomes".(P14, line 406-430).

[1]Hastie, P. A., & Curtner-Smith, M. D. (2006). Influence of a hybrid Sport Education—Teaching Games for Understanding unit on one teacher and his students. Physical Education and Sport Pedagogy, 11(1), 1-27. DOI: 10.1080/17408980500466813

P22, L402-406: Why here again the theory of the model is explained. It is about discussing, not expounding theory that should be part of the theoretical framework. In this case of TPSR.

Thank you for your insightful comment. We agree that the discussion section should focus on interpreting and analyzing the results rather than reiterating theoretical explanations. In response to your feedback, we have moved the theoretical details to the case of TPSR as part of the theoretical framework (P3, Line 82-87).

The article would contribute more if it showed how to implement these hybridisations.

Thank you for your valuable feedback. We agree that demonstrating how to implement these hybrid models would be beneficial. In the Introduction section (P3, line 111-122), we have described how SEM can be combined with other teaching models and provided examples.

However, our review of the literature indicates that there is currently a lack of detailed descriptions for a fixed or optimal method of implementing the "SEM + 1" model. The approach to mixing teaching models often depends on the specific objectives of the instructor, reflecting the inherent flexibility of the "SEM + 1" model. Different environments and goals may require different implementation methods.

Additionally, in Section “3.3 Hybrid Curriculum Implementation” of the Results, we emphasize that the learning tasks and teaching content of hybrid models primarily derive from the integration of SEM with another partner model. To enhance the contribution of the article based on your suggestion, we have also added a new section on "4.1 Hybrid Curriculum Implementation" in the Discussion (P13-14, line 387-403), where we further address these issues.

We appreciate your understanding and hope this explanation clarifies our approach.

4.7 Implementation limitations and future applications

Can they indicate why you think that L454 “resulting in the teaching content of the other mode of teaching is often interrupted”?

Thank you for the reviewer’s insightful comment. Our original statement, “resulting in the teaching content of the other mode of teaching is often interrupted,” was based on the study by Hastie & Curtner-Smith (2006). In the discussion section under the heading “Issues, problems, and challenges,” they stated: “Although this unit was lengthy when compared to traditional multi-activity units, it was still difficult to attend to all the instructional and managerial tasks associated with both SE and TGfU. […] Within a 30-minute lesson, of course, this meant that TGfU-style instruction tended to get squeezed out.” Our understanding at that time was that when combining these two teaching models, the content of one mode could potentially be interrupted. However, upon reflection, we realize this wording might be somewhat misleading, so we have revised the text to better align with our study's findings and intentions. (P15-16, Line 484-489)

Please, review long sentences: example L450-454.

Thank you very much for the suggestion. We have thoroughly reviewed the manuscript for long sentences and have simplified them for clearer expression throughout the text. (P15-16, Line 484-489)

Conclusion

Please review the 1st sentence of the conclusion “This systematic review systematically examines”

Thank you for your suggestion. We have revised the first sentence of the conclusion to improve clarity.

In general, the conclusions do not respond to the aims of the review.

Thank you for your feedback. We have revised the conclusion to better align with the aims of the review. (P16, Line 500-509)

References

Please check the dois in all the references, in Shen's reference does not work.

Thank you very much for your meticulous review. We sincerely apologize for the errors in our writing, and they have now been corrected. Additionally, we have carefully checked and verified the DOIs for all references listed.

Finally, we would like to once again express our gratitude for your guidance in helping us improve the clarity and accuracy of the paper. We look forward to your further comments.

Reviewer #2: 

The author(s) did a good review which shows that they understand the concept that is been review. I am recommending that the manuscript be accepted for publication. The author(s) should take second look at the the language and ensure grammatical errors are corrected.

Thank you very much for your positive feedback and recommendation for the acceptance of our manuscript. We truly appreciate your acknowledgment of our understanding of the reviewed concept. In response to your suggestion, we have carefully reviewed the language throughout the manuscript and corrected any grammatical errors. We hope that the revised version meets your expectations, and we would like to express our sincere gratitude once again for your contribution to the publication process.

---

## [Decision Letter · Decision Letter 1]

30 Sep 2024

Optimizing learning outcomes in Physical Education: A comprehensive systematic review of hybrid pedagogical models integrated with the Sport Education Model

PONE-D-24-20221R1

Dear Dr. zhang,

We’re pleased to inform you that your manuscript has been judged scientifically suitable for publication and will be formally accepted for publication once it meets all outstanding technical requirements.

Kind regards,

Musa Adekunle Ayanwale

Academic Editor

PLOS ONE

Additional Editor Comments (optional):

Thank you for your detailed and well-organized responses to the reviewers' comments. I appreciate your efforts in addressing each point raised, which have significantly improved the quality and clarity of your manuscript. The reviewers have acknowledged your thoughtful revisions, and Reviewer #2 has recommended acceptance of the manuscript. Reviewer #1’s concerns regarding the theoretical framework, references, methodology, and implementation details have been comprehensively addressed, and the updates have enhanced the manuscript’s contribution to the field.

Based on the positive feedback and your revisions, I am pleased to inform you that the manuscript is accepted for publication with no further revisions needed.

Congratulations, and thank you for your contribution!

Musa Adekunle Ayanwale

Academic Editor

PLOS ONE

Reviewers' comments:

Reviewer's Responses to Questions

**Comments to the Author**

1. If the authors have adequately addressed your comments raised in a previous round of review and you feel that this manuscript is now acceptable for publication, you may indicate that here to bypass the “Comments to the Author” section, enter your conflict of interest statement in the “Confidential to Editor” section, and submit your "Accept" recommendation.

Reviewer #2: All comments have been addressed

2. Is the manuscript technically sound, and do the data support the conclusions?

Reviewer #2: Yes

3. Has the statistical analysis been performed appropriately and rigorously? 

Reviewer #2: N/A

4. Have the authors made all data underlying the findings in their manuscript fully available?

Reviewer #2: Yes

5. Is the manuscript presented in an intelligible fashion and written in standard English?

Reviewer #2: Yes

6. Review Comments to the Author

Reviewer #2: The auhor(s) did a good job and have shown mastery of the article submitted for publication. I approved the article f

7. PLOS authors have the option to publish the peer review history of their article (what does this mean?). If published, this will include your full peer review and any attached files.

Reviewer #2: **Yes: **Udeme Samuel Jacob

---

## [Editor Report · Acceptance letter]

11 Oct 2024

PONE-D-24-20221R1 

PLOS ONE

Dear Dr. zhang, 

I'm pleased to inform you that your manuscript has been deemed suitable for publication in PLOS ONE. Congratulations! Your manuscript is now being handed over to our production team.

Kind regards, 

on behalf of

Dr. Musa Adekunle Ayanwale 

Academic Editor

PLOS ONE